# How to Combat Over-Testing for Patients Suspected of Pulmonary Embolism: A Narrative Review

**DOI:** 10.3390/diagnostics13071326

**Published:** 2023-04-03

**Authors:** Pierre-Marie Roy, Thomas Moumneh, Thomas Bizouard, Damien Duval, Delphine Douillet

**Affiliations:** 1Department of Emergency Medicine, University Hospital of Angers, Avenue of the Hotel Dieu, 49100 Angers, France; 2UMR MitoVasc CNRS 6215 INSERM 1083, University Angers, 49100 Angers, France; 3FCRIN, INNOVTE, 42023 Saint-Étienne, France; 4Department of Emergency Medicine, University Hospital of Tours, Avenue of the Republic, 37044 Tours, France

**Keywords:** over-testing, pulmonary embolism, diagnostic

## Abstract

The diagnosis of PE remains difficult in 2023 because the signs and symptoms are not sensible nor specific. The consequences of potential diagnostic errors can be dramatic, whether by default or by excess. Furthermore, the achievement of a simple diagnostic strategy, based on clinical probability assessment, D-dimer measurement and computed tomography pulmonary angiography (CTPA) leads to a new challenge for PE diagnosis: over-testing. Indeed, since the 2000s, the wide availability of CTPA resulted in a major increase in investigations with a mod I confirm erate increase in PE diagnosis, without any notable improvement in patient outcomes. Quite the contrary, the complications of anticoagulation for PE increased significantly, and the long-term consequences of imaging diagnostic radiation is an important concern, especially the risk of breast cancer for young women. As a result, several strategies have been proposed to fight over-testing. They are mostly based on defining a subgroup of patients for whom no specific exam should be required to rule-out PE and adjusting the D-dimer cutoff to allow the exclusion of PE without performing CTPA. This narrative review presents the advantages and limitations of these different strategies as well as the perspective in PE diagnosis.

## 1. Introduction

Pulmonary embolism (PE) is a common and serious condition. Its reported incidence in industrialized countries ranges from 0.3 to 0.8 per 1000 inhabitants, i.e., more than 150,000 new cases per year in the USA [1,2,3]. The clinical signs are neither very sensitive nor very specific, which is why the diagnosis is often put forward as a possibility. Although only one PE is diagnosed for around every 10 to 20 suspected cases, more than one million diagnostic approaches are initiated every year in America and Europe to screen for a PE [4,5,6,7].

The quality of these diagnostic approaches and the resulting decisions are crucial. The consequences of potential diagnostic errors can be dramatic, whether by default or by excess. Without anticoagulant treatment, around 50% of cases of pulmonary embolism recur within 3 months, and 10% of cases of symptomatic PE are lethal [8]. The risk of death due to PE or thromboembolic recurrence is 27 times higher without anticoagulant treatment compared to when treatment is started immediately after the diagnosis [9]. In 2004, of 370,000 deaths linked to thromboembolic disease in six European countries, 59% of deaths corresponded to undiagnosed PEs [10]. A very recent analysis of misdiagnosis in the Emergency Departments (ED) commissioned by the Agency for Healthcare Research and Quality revealed that vascular events was the first condition of the seven million diagnostic errors occurring each year in US ED [11]. On the other side, among patients under anticoagulant treatment, the rate of major hemorrhaging was 13.4% per year, with 1.15% of intracranial hemorrhaging in the year 2000 [12]. In the three months after starting anticoagulant treatment for a thromboembolic episode, 1.6% of patients presented with major bleeding, including 0.2% with fatal bleeding; 3.4% presented with thromboembolic recurrence despite treatment, including 0.4% for which the PE recurrence was fatal [13]. The risk of major bleeding and the risk of death linked to initial thromboembolic disease or its recurrence is significantly higher in patients with PE compared to patients with venous thrombosis without PE [9]. Taking into account only hemodynamically stable patients with PE under anticoagulant treatment, the rate of death due to PE is 3.1% at 1 month, the rate of serious hemorrhage is 2.2% and the rate of non-fatal recurrence of PE is 0.6% [14,15]. Fortunately, in recent years, fatal PEs and severe bleeding have tended to decrease, due to the introduction of direct oral anticoagulants [9].

In the 2000s, as a result of introducing the D-dimer test and CT pulmonary angiography (CTPA) and drawing on Bayes’ theorem, it was possible to validate a simple diagnostic approach that can be used in most cases at ED in industrialized countries. The approach is first and foremost based on an estimated clinical probability of PE, which helps determine in which prevalence group the patient is in: low, moderate or high. If the clinical probability is not high, a D-dimer test is performed, which allows one to rule out PE if it is negative (<500 μg/L) with a very sensitive quantitative technique (the ELFA method derived from the ELISA methodology or the latex turbidimetry method). If clinical probability is high or if the D-dimer test is positive, a CTPA is required. An alternative to the CT scan—as in patients with severe kidney failure or a confirmed allergy to contrast media—is performing a ventilation–perfusion pulmonary scintigraphy. Several multi-center studies have shown the latter to be reliable [16,17,18], and it is included in national and international recommendations [19,20,21]. Pulmonary angiography is an invasive examination. It is performed by injecting intravenous contrast directly into the pulmonary arteries via a percutaneous catheter introduced into the heart under fluoroscopic guidance. This examination was associated with a 6.5% rate of adverse events. Although described as the old gold standard for the diagnosis of PE, pulmonary angiography is now rarely used for the diagnosis of PE. It has been shown that its diagnostic performance is similar to that of CT pulmonary angiography, being less invasive and easier to perform. 

## 2. Over-Testing

The achievement of this simple diagnostic approach leads to a new paradigm for PE suspicion: over-testing and over-diagnosis [22]. The concept of over-testing refers to the benefit–risk balance of diagnostic investigations. An over-testing is established when the risk related to the investigations and treatment of underlying diagnosed PE exceeds the benefit of diagnosing PE, i.e., the risk leaving untreated undiagnosed PE. 

In the last decades, the number of patients who were investigated for PE suspicion and who underwent a CTPA was multiplied by ten. The analysis of the prevalence of PE among suspected patients for which a CTPA approach was initiated dropped from about 25% in the 2000s to around 5% in the most recent trials, meaning that 20 patients must be investigated to diagnose one PE [4,5,6,7,16,23,24,25,26,27,28,29]. The rate of adverse events related to diagnostic investigations for PE is globally low. Nevertheless, with regard to the number of patients investigated, the risks of CTPA need to be formally assessed. According to some authors, the risk of kidney function deteriorating while performing a CTPA is 14% [30]. It is, however, debated whether this alteration is related to contrast media (induced nephropathy). A recent meta-analysis did not show a significant difference regarding the rate of kidney failure between patients who underwent a CT scan with and without injection of a contrast agent [31]. Radiation linked to thoracic CT scans is very low, from 6 to 10 mSV, well below the risk of fetal toxicity in pregnant women. Nevertheless, most of the radiation is focused on the lungs and breasts. The increased risk of neoplasm induced by radiation linked to a CTPA could reach 0.61% for bronchopulmonary cancer and 0.4% for breast cancer in 20-year-old women [32]. Repeated diagnostic radiology exams, especially CT scans, could, therefore, be responsible for a significant share of induced cancers [33]. Moreover, CTPA may lead to incidental findings, i.e., findings not supporting PE or another alternative diagnosis and unrelated to the acute symptoms of the patients. In prospective studies, this may concern up to 50% of patients, some patients having several incidental findings [34,35]. More than half of them are minor and do not require any follow-up. However, incidental findings leading to clinical or radiological follow-up are observed in near 25% of patients. How these discoveries are helpful and beneficial for patients or unnecessarily stressful is unknown.

The main risk of over-investigation is over-diagnosis, however. Over-diagnosis refers to putting forward an incorrect diagnosis and/or diagnosing a PE that has no clinical significance and for which initiating anticoagulant treatment is more dangerous than it is helpful [36]. The greater the use of CTPA, the greater the risk of false positives, especially when the quality of the CTPA is suboptimal. A retrospective analysis by expert radiologists of CTPAs performed over 10 years in an emergency department revealed up to 26% of over-diagnosis errors, mainly when the diagnosis related to a single embolism image and/or segmental or subsegmental defects [37]. As the resolution in CTPAs improved with the introduction of multi-slice CT scanner technology, subsegmental PE diagnoses became significantly more common compared to diagnostic strategies based on first-generation CTPAs or pulmonary scintigraphy, although the risk of symptomatic thromboembolic events during follow-up did not change [26,38]. A significant share of subsegmental PEs, therefore, may not justify introducing anticoagulant treatment (especially in patients under 65 years of age and patients with an isolated single sub-segmental PE) [39,40].

Based on the main diagnoses during hospitalization in the United States, Wiener et al. showed that the incidence of PEs, which was stable from 1993 to 1998, increased by 80% between 1999 and 2006 (from 62/100,000 to 112/100,000) [41]. Of note, the incidence rate of DVT alone remained relatively stable for this period confirming that the observed increase in PE incidence was not related to an overall increase in thrombosis events but linked to the advances in PE diagnosis [3]. Conversely, mortality linked to PE continued to drop but to a lesser extent than before CTPA were introduced (−8% versus −3%) and to a much lesser extent than the rise in PE diagnoses. Moreover, the presumed in-hospital complications of anticoagulation for PE increased significantly (+71%) [41]. Limited evidence exists for the diagnosis of recurrent venous thromboembolic events. These patients are at risk of over-testing: clinical decision rules are less useful and D-dimer performance is reduced. The efficiency of the algorithm is relatively low compared to patients without a history of VTE [42].

In total, the diagnostic strategy based on a D-dimer test and a CTPA has made the diagnostic approach significantly easier when PE is suspected and has made the exclusion strategy more reliable but without a significant effect on mortality linked to PE and by increasing the risk of over-testing and over-diagnosing [43]. As such, it has become a priority to introduce a de-escalation approach that limits the use of imaging tests without increasing the risk of errors by default [44].

## 3. The Strategies Designed to Limit the Use of Imaging Tests

### 3.1. The PERC Rule

The first approach to limit the use of imaging was to be more accurate in determining which patients should undergo a diagnostic approach to screen for PE. Due to the low sensitivity of simple clinical and paraclinical signs, the approach highlighted in most studies and scientific recommendations is to put forward a PE diagnosis in any patient who presents relevant symptoms, especially recent dyspnea and/or chest pain if no other diagnostic explanation has been formally established [21,45]. This approach means investigating a large number of patients, including cases where the physician is convinced that the patient is not experiencing a PE. As this is especially true in the United States due to medicolegal pressure, Jeffrey Kline’s team developed the PERC rule, i.e., the Pulmonary Embolism Rule-out Criteria. The objective of the PERC rule is to rule out PE based on clinical data alone, without any specific paraclinical investigations, including a D-dimer test (Table 1). The rule was devised using a derivation cohort of 3148 patients with suspected PE and validated on a cohort of 1427 patients with a low suspicion of PE according to the implicit assessment of the emergency department physician (8% prevalence) with a false-negative rate of 1.4% (95% CI: 0.4–3.2) [46]. This strategy was then prospectively evaluated in several large cohorts of patients, mainly in the US [47,48]. Around 20% of patients had a low clinical probability evaluated implicitly by the physician (estimated < 15%) and a negative PERC rule and were not investigated to screen for PE. During the 45 days of follow-up, 1% (95% CI: 0.6–1.6) presented with a symptomatic thromboembolic event considered as a possible false negative to the rule [49].

In Europe, retrospective studies performed in populations where the prevalence of PE was higher turned out to be less reassuring, with a false-negative rate of more than 5%, including when the PERC rule was associated with a low clinical probability according to the revised Geneva score (Table 1) [50,51]. Nevertheless, more recent studies have confirmed the reliability of a negative PERC rule if applied in patients with a low implicit clinical probability [4,50]. An implementation cluster randomized trial in France showed that using the PERC rule in patients with a low implicit clinical probability was safe and helped reduce the rate of CT angiography use from 23% to 13% [7]. Therefore, the PERC rule is included in American, European and French recommendations, specifying that it must be used only in the case of patients with a low clinical probability evaluated implicitly by the physician [21,45,52,53].

#### Limitations

The PERC rule requires a prior assessment of implicit clinical probability and can only be applied in populations where prevalence is low. In the PROPER study, which has shown to be beneficial in France, the prevalence of PE was 2.7% in the control group and 1.5% in the intervention group [7]. In practice, it can only be used when the physician—after analyzing simple clinical and paraclinical data—believes that it is unlikely or very unlikely that the patient is experiencing a PE [21]. This evaluation cannot be made using a clinical probability score such as the revised Geneva score, a negative PERC rule having an insufficient negative predictive value with a false-positive rate of >5% [50,54]. Moreover, it cannot be used in pregnant women or in the postpartum period [4,55]. 

### 3.2. Age-Adjusted D-Dimer Cut-Off Level

The second approach involves optimizing the D-dimer test. Taking 500 μg/L as the positivity cut-off level, the D-dimer test performed using very sensitive quantitative techniques has a very good exclusion value [17,56,57,58,59,60]. However, its low specificity at around 40% means that many examinations do not provide any relevant information because they are above this exclusion value [59]. This is especially true in elderly patients seeing as D-dimer levels increase physiologically with age [61,62]. Several retrospective studies have shown that an age-adjusted cut-off level in patients over 50 years old helps increase the rate of “negative” D-dimers without any loss of reliability [63,64,65,66]. With a very sensitive quantitative method (mainly ELFA), an assay below 500 μg/L before the age of 50 years or lower than [age × 10] μg/L after the age of 50 years is considered negative [66]. This approach has been validated in an international multi-center study published in 2016: ADJUST-PE [67]. In patients with a low or moderate clinical probability according to the revised Geneva score or “not very likely” according to the two-level Wells score, a negative age-adjusted D-dimer test makes it possible to rule out PE (Table 1). Among the 3346 patients included in the study, 331 (10%) had a D-dimer test higher than 500 μg/L but lower than [age × 10] μg/L, and one of them experienced a thromboembolic event during follow-up: 0.3% [95% CI: 0.1–1.7%]. In elderly patients over 75 years old, the rate of patients for which PE could be ruled out without using thoracic imaging was multiplied by five (from 6.4% to 29.7%) [67]. 

#### Limitations

The age-adjusted D-dimer cut-off value has been validated in the ADJUST study, with an evaluation of clinical probability according to the revised Geneva score or the two-level Wells score [67]. Used in isolation, its impact on reducing the use of CT scans is modest because it does not apply to young patients with a low level of suspicion who are the most common type of patients at the emergency department and for whom the question of iatrogenic injury linked to radiation is most relevant [67].

### 3.3. D-Dimer Cut-Off Level Adjusted to Clinical Probability

Another approach to optimizing the D-dimer test has involved taking a cut-off value that varies depending on the level of clinical probability. Based on Bayes’s theorem, the exclusion value of a test depends on its intrinsic performance (evaluated using its negative likelihood ratio (NLR), a parameter that combines sensitivity and specificity) and on the prevalence of the condition in the population where the test is applied, also called the pre-test probability (evaluated using clinical probability) [68]. For the same NLR, the lower the clinical probability, the lower the risk of a false negative and vice versa [69,70,71]. This explains that the exclusion value for a D-dimer test < 500 μg/L or age-adjusted D-dimer test (NLR 0.01) is poor in the event of high clinical probability. Conversely, in the event of low clinical probability, a less accurate test can turn out to be sufficient to rule out PE more reliably [17]. Several authors have evaluated the reliability and benefits of using a higher cut-off value (losing in sensitivity and exclusion value but gaining in specificity) in the event of low clinical probability and, potentially, a lower cut-off value in the event of high probability [72,73,74,75]. 

In an observational study, Kline et al. showed that using a cut-off value < 1000 μg/L in the event of low clinical probability patients according to the revised Geneva score or the Wells score could help significantly reduce the use of thoracic CT angiography in exchange for a moderate increase in false negatives (*n* = 11/208, 5.8%), which are essentially isolated cases of subsegmental PE of an undetermined clinical significance (*n* = 10/11, 90.9%) [76]. In 2017, Van der Hulle et al. validated this approach in a prospective multi-center pragmatic interventional study on 3465 patients with suspected PE at the emergency department [29]. Clinical probability was evaluated using the YEARS rule based on three items of the Wells score: signs of deep vein thrombosis, hemoptysis and PE being the most likely diagnosis (Table 1). The lack of these three criteria needed to be confirmed for the clinical probability to be low, i.e., to consider the YEARS rule as negative. All patients underwent a D-dimer test. When the YEARS rule was negative, the cut-off value < 1000 μg/L was applied, and PE was ruled out. When the rule was positive (not low clinical probability), the cut-off value < 500 μg/L was applied. Assays higher than the cut-off value required performing a thoracic CT angiography. In their cohort for which prevalence was at 13%, the authors showed that this strategy made it possible to rule out PE without using CT scanning in 46% of cases with, overall, a very low risk of false negatives (thromboembolic events during the 3 months of follow-up: 0.61% (95% CI: 0.36–0.96%)). If the algorithm based on the Wells score and an age-adjusted D-dimer test had been applied, the rate of patients with no indication of CT scanning would have been 39%. Among the 1473 patients with a negative YEARS rule, 1320 had D-dimer levels of <1000 μg/L, including 1302 who had not received anticoagulant treatment and who were followed up at 3 months. Among them, six suffered a thromboembolic event during the follow-up period (0.5%). Notably, the YEARS strategy was prospectively assessed in a cohort of 498 pregnant women suspected of PE with a PE prevalence of 4%. The failure rate was 0.21% (95% CI: 0.04–1.2%). CTPA was avoided in 65%, 46% and 32% of the patients investigated during the first trimester, the second trimester and the third trimester, respectively [77].

Recently, the Kearon et al. team evaluated a very similar strategy by drawing on the three-level Wells score combined with a slightly modified probability stratification: the PEGeD study (Table 1) [5]. When the Wells score was lower than or equal to 4 (and not ≤2), clinical probability was considered low. In such cases, a D-dimer test was recommended, using <1000 μg/L as the cut-off value that ruled out PE. When the Wells score was between 4.5 and 6, clinical probability was moderate and a D-dimer test was recommended, with <500 μg/L as the exclusion value. When the Wells score was >6, clinical probability was considered to be high, and imaging was recommended. The prevalence in this study, which included 2017 patients, was 7.4%. The false negatives rate in the entire cohort was 1/1863: 0.5% (95% CI: 0.01–0.030%) and, in the low probability group, 0% (95% CI: 0.00–0.29%) [5]. The strategy was, therefore, confirmed to be reliable.

#### Limitations

The YEARS rule is based on clinical probability and on a D-dimer test for all patients [29]. As such, it does not include determining a subgroup of patients with a high probability and for which the negative predictive value of a D-dimer test of <500 μg/L is insufficient to rule out PE without any doubt. By considering that the maximum acceptable risk of a false negative is 1.9% and that the negative likelihood ratio of a D-dimer test is 0.01, the YEARS rule should not be applied if the pre-test probability is high with a prevalence over 65% [17,18,19,20,21,22,23,24,25,26,27,28,29,30,31,32,33,34,35,36,37,38,39,40,41,42,43,44,45,46,47,48,49,50,51,52,53,54,55,56,57,58,59,60]. In the YEARS study, the physicians performed a CTPA in 24 patients despite a D-dimer test of <500 μg/L, most likely because they considered clinical probability to be high. Among them, three patients experienced a PE: 12.5% [29]. The PEGeD study is based on the Wells score with an adaptation of cut-off values, with clinical probability considered low when the score is <4, i.e., the same value as that used in the ADJUST study for considering PE not very likely. An external validation of the PEGeD strategy on an independent cohort with a higher prevalence of PE (22%) showed less favorable results. The overall failure rate of the strategy was 52/2610: 2% (95% CI: 1.5–2.6%). In the subgroup of patients with a D-dimer below 1000 μg/L but above the age-adjusted cutoff, it was 36/414: 8.7% (95% CI: 6.4–11.8%) [78]. These results confirm the importance of the prevalence of PE and the way in that the pretest probability is assessed.

### 3.4. Combined Strategy

Combining these different strategies may be the more efficient way to reduce over-testing if it is proven to be safe. In a retrospective study involving 1951 patients who overall had a low probability (3.9% prevalence), combining the PERC rule and the YEARS rule made it possible to significantly reduce the number of CTAs, with a low rate of false negatives: 0.83% (95% CI = 0.51–1.35) [79]. Freund et al. assessed this possibility in a cluster randomized cross-over trial in 18 ED [6]. The first step was the assessment of implicit clinical probability (CP) of PE by the physician in charge, and the second step was the assessment of the PERC rule. Patients with a high CP were excluded as well as patients with a low CP and a negative PERC rule. The other patients suspected of PE were included and have a D-dimer test. In the intervention period, the YEARS rule was assessed. For patients with a negative YEARS rule (none of the YEARS items), 1000 μg/L was applied as the D-dimer cutoff; for others, an age-adjusted cutoff was applied. In the control period, an age-adjusted cutoff was applied for all patients. In both group, patients with a D-dimer test above the cutoff value had a CTPA. The trial included 1414 patients, 726 in the intervention group (PE prevalence 7.4%) and 688 in the control group (PE prevalence 6.7%). In the per-protocol population, the failure rate was 0.15% (95% CI 0.00% to 0.86%) and 0.80% (95% CI 0.26 to 0.86%) in the intervention group and the control group, respectively, with confirmation of non-inferiority between groups. Chest imaging was performed in 30.4% and 40.0% in the intervention group and the control group, respectively [6].

#### Limitations

The main limitation of this combined strategy is that it required assessing clinical probability in three different ways: implicit evaluation, followed by an evaluation according to the PERC rule, followed by an evaluation according to the YEARS rule. It could make it complicated to implement in current practice, all the more so because the strategy does not apply to high clinical probability. There is a risk of confusion between these different methods. For example, if the PERC rule is applied together with an estimate of clinical probability according to the revised Geneva score, the risk of error by default—i.e., of false negatives—is higher than 5% [50,54]. 

### 3.5. PEPS Strategy

The 4PEPS, or the 4-Level Pulmonary Embolism Clinical Probability Score, was designed in order to safely limit the use of imaging in the case of suspected PE in the emergency department by using, as part of one score, the different strategies developed previously (Table 2) [80]. Based on ISTH recommendations and depending on the prevalence of VTE, the acceptable number of false negatives has been set to 2% [81,82]. The target prevalence values in each category of pre-test clinical probability (before resorting to additional examinations) have been calculated using Bayes’s theorem based on negative likelihood ratios and the objective of the post-test probability (≤2%): [71] 

-Very low CP: ≤2%, which makes it possible to rule out PE without performing any additional examinations, including the D-dimer test.-Low CP: ≤20%, which makes it possible to rule out PE based on a D-dimer test with a very sensitive technique < 1000 μg/L (D-dimer NLR < 1000 µg/L: 0.08).-Moderate CP: ≤65%, which makes it possible to rule out PE based on a D-dimer test with an age-adjusted cut-off value after the age of 50 years (age-adjusted D-dimer NLR: 0.01).-High CP: >65%, which does not make it possible to rule out PE based on a D-dimer test and requires the use of thoracic imaging straight away.

The 4PEPS score was initially derived from and validated based on databases from three studies, which in total included 11,066 patients with suspected PE [16,49]. The overall prevalence in all the databases was 11%. It was then validated in the internal validation cohort and in two external validation cohorts, one with an overall prevalence of 21% (*n* = 1546) and another with a prevalence of 11.7% (*n* = 1669) [4,18].

The area under the ROC (receiving operating curve) in the validation cohorts was, respectively, 0.82 (95% CI: 0.83–0.86) for the internal validation cohort, 0.79 (0.76–0.82) in the first external validation cohort and 0.77 (0.74–0.80) in the second external validation cohort. This means that 4PEPS compares very favorably with other existing scores [80,83,84]. If the 4PEPS strategy had been applied in the two external validation cohorts (retrospective estimate), the percentage of false negatives would have been *n* = 11/1548, 0.71% (95% CI: 0.37–1.23) and *n* = 14/1570, 0.89% (95% CI: 0.53–1.49), respectively. This rate is in line with prerequisites with an upper limit of the confidence interval < 1.93% and 1.88%, respectively, (value established depending on the prevalence of PE), and it is similar to the rates observed in recent studies that have evaluated applying the PERC rule, the YEARS rule or the PEGeD strategy [4,5,29,85]. Conversely, the rate of CT scan use dropped by 22% (46% vs. 68%) and by 19% (32% vs. 51%) compared to the standard approach. It also significantly dropped compared to other strategies applied retrospectively in the external validation cohorts [80]. Two other recent studies evaluated the efficacy and safety of the 4PEPS strategy. Compared retrospectively to the YEARS rule, the 4PEPS strategy helped avoid 10% more CT scans (58% versus 48%) while maintaining an acceptable rate of false negatives according to international recommendations, namely 1.3% (95% CI: 0.86–1.9). In this study, the performance of 4PEPS as measured by the area under the ROC was good, with an AUC of 0.82 (95% CI: 0.80–0.84) [86]. Roussel et al. compared different strategies retrospectively based on patient data from the Modigliani study and the PROPER study: PERC + YEARS, PERC + PEGeD and 4PEPS [87]. In this population of 3330 patients with an overall low clinical probability (overall prevalence of 4.5%), no difference was found between these three strategies in regard to the failure rate with 0.90% (0.63% to 1.28%) for the 4PEPS strategy. The 4PEPS strategy leads to the lowest proportion of imaging testing: 14.9% [87].

#### Limitations

The main limitation of the 4PEPS strategy is that it requires a formal assessment in a prospective implementation trial. Such trial is planned and should start in 2023 [88]. Furthermore, 4PEPS contains 12 items and is complex to memorize. It requires, therefore, the use of a calculator or computerized help for decision making. As with other strategies except YEARS, the 4PEPS does not apply to pregnant women.

## 4. Conclusions and Perspectives

Several strategies have been developed to safely limit over-testing. The most promising combined several possibilities: (1) defining a subgroup of patients with a very low probability of PE for whom no specific exam should be required to rule-out PE (as the PERC strategy), (2) a subgroup of patients with a low probability of PE for whom a D-dimer cut-off of 1000 μg/L can be applied (as the YEARS strategy or the PEGeD strategy) and (3) patients for whom an age-adjusted cut-off for D-dimer should be applied (as the ADJUST strategy).

However, the original strategies used different methods to assess clinical probability and their combination may be difficult to implement in everyday clinical practice. Finally, two options emerge. The first one is to define a strategy as simple as possible and mainly based on implicit assessment. Such a Modified Simplified (MODS) strategy with YEARS reduced to the sole item of “Is PE the most likely diagnosis?” combined with the PERC rule was recently proposed. It exhibited a failure rate similar to other strategies in a retrospective study [87]. A similar strategy named the Adjust-Unlikely algorithm will be assessed soon in a pilot prospective trial in Canada (NCT05708794). The other option is to implement a more complex strategy using a computerized help decision-support system (CDSS) available on physicians’ smartphone or computers. In the SPEED&PEPS trial assessing the safety and efficacy of the 4PEPS strategy as compared to the current practice, the 4PEPS is included in the CDSS called SPEED “Suspected Pulmonary Embolism in Emergency Department” [88]. Such tools are also considered a means of fighting misdiagnosis in emergency departments [11]. 

In conclusion, solutions exist to deal with over-testing for PE even if combined strategies still need to be prospectively evaluated. Nevertheless, the most important challenge is to implement them in everyday clinical practice.

## Figures and Tables

**Table 1 diagnostics-13-01326-t001:** Clinical rules and scores.

PERC Rule		Revised Geneva Score		Wells Score		YEARS Rule	
Age ≥ 50 years	+1	Age > 65 years	+1	PE = the most likely diagnosis	+3	PE = the most likely diagnosis	+1
Personal history of VTE	+1	Personal history of VTE	+3	Personal history of VTE	+1.5		
Heart rate ≥ 100/min	+1	Heart rate between 75 and 94/min	+3	Heart rate > 100/min	+1.5		
		Heart rate ≥ 95	+5				
Hemoptysis	+1	Hemoptysis	+2	Hemoptysis	+1	Hemoptysis	+1
Recent trauma or surgery (≤4 weeks)	+1	Recent fracture or surgery (≤4 weeks)	+2	Recent immobilization or surgery (≤4 weeks)	+1.5		
Hormone treatment	+1	Active cancer or cancer in remission < 1 year	+2	Active cancer or cancer in remission < 1 year	+1		
		Unilateral leg pain	+3				
Edema of a lower limb	+1	Clinical signs of DVT	+4	Clinical signs of DVT	+3	Clinical signs of DVT	+1
Negative rule = 0 Positive rule ≥ 1		Low probability ≤ 3 Moderate 4–10 High ≥ 11		Low (PE not likely) ≤ 4 Moderate 4.5–6 High ≥ 6.5		Negative rule = 0 Positive rule ≥ 1	

**Table 2 diagnostics-13-01326-t002:** 4PEPS or 4-Level Pulmonary Embolism Clinical Probability Score.

Field	Criterion	Points
Demographic data	Age < 50 years old	−2
	Age ≥ 50 years old and <65 years old	−1
	Male	+2
Medical history	Chronic pulmonary disease	−1
	Personal history of VTE	+2
	Estrogen hormone therapy	+2
	Bed rest, surgery or orthopedic immobilization	+2
Signs and symptoms	Heart rate < 80/min	−1
	Combination of chest pain and acute dyspnea	+1
	Syncope	+2
	Oxygen saturation (SpO2) < 95%	+3
	Signs of thrombosis (calf pain and/or edema)	+3
Differential diagnosis	PE is the most likely diagnosis	+5
Categories of clinical probability	Very low CP	<0
	Low CP	0 to 5
	Moderate CP	6 to 12
	High CP	>12

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
