# Peer review of "How to Combat Over-Testing for Patients Suspected of Pulmonary Embolism: A Narrative Review"

_diagnostics, 2023, doi:10.3390/diagnostics13071326_

Round 1

Reviewer 1 Report

My main objection against the publication of this review is the fact that I do not evaluate the diagnostic procedures associated with the suspicion of PE as the over-testing. Expert authors repeatedly confirmed that the risk of breast cancer was 7.0–7.1 per 10,000 women and did not differ between those exposed to CTPA, V/Q scanning and unexposed women. Both scanning techniques were below the safety threshold of 100mGy, associated with fetal radiation complications.

Moreover, on the other hand, I think that it is better to take the PE into account in the differential diagnosis of the patient´s complaints and to exclude it rather than to overlook it and to cause the uninevitable death of the patient – it is better to come to negative conclusions with the patient who is alive that to contribute to the patient´s death doing nothing.

Based on the arguements described above, I recommend the rejection of the manuscript.

Author Response

We understand your argument. Indeed, failure to diagnose a pulmonary embolism can have important consequences in terms of morbidity-mortality for the individual patient. However, at the collective level, it is important to question our practices and the relevance of diagnostic strategies. An attempt to rationalize the diagnostic process seems necessary, which is shared by the vast majority of authors (Kahn et al, N Engl J Med 2022, Freund et al, JAMA 2022, Dobler et al, Breathe (Sheff), 2019). Moreover, this fight against over-testing is done safely, as shown by the low failure rate of the new strategies. For the YEARS, of the 2946 patients (85%) in whom pulmonary embolism was ruled out at baseline and remained untreated, 18 patients were diagnosed with symptomatic venous thromboembolism during 3-month follow-up (0.61%, 95% CI 0.36–0.96)(van der Hulle et al, Lancet, 2017). For the age adjusted D-dimer, the 3-month failure rate in patients with a D-dimer level higher than 500 μg/L but below the age-adjusted cutoff was 1 of 331 patients (0.3% [95% CI, 0.1%-1.7%])(Righini et al, JAMA 2014). The strategies put in place to combat overtesting make it possible to avoid making diagnoses with little or no clinical impact and to avoid missing potentially severe PE.

Best regards, 

Prof Pierre-M Roy et Dr Delphine Douillet

Reviewer 2 Report

Greetings,

I read your manuscript on strategies to prevent excessive tests in the diagnosis of pulmonary embolism with great interest. Overall, I believe the manuscript is of good quality for publication. I have only a few comments on the manuscript:

-In Table 1, did the authors mean clinical signs of thrombosis due to deep vein thrombosis (DVT) instead of "Clinical signs of thrombosis due to MI"?

-Please add the role of pulmonary angiography to the manuscript.

-Please reduce self-citations, if possible, unless it is absolutely necessary to use previous publications

Author Response

-In Table 1, did the authors mean clinical signs of thrombosis due to deep vein thrombosis (DVT) instead of "Clinical signs of thrombosis due to MI"?

We thank reviewer 2 for this comment. We have corrected it as suggested.

-Please add the role of pulmonary angiography to the manuscript.

We have added some elements explaining the preference to use CTPA and not pulmonary angiography.

“Pulmonary angiography is an invasive examination. It is performed by injecting intravenous contrast directly into the pulmonary arteries via a percutaneous catheter introduced into the heart under fluoroscopic guidance. This examination was associated with a 6.5% rate of adverse events (Stein et al Circulation, 1992). Although described as the old gold standard for the diagnosis of PE some decades ago, pulmonary angiography is today exceptionally used for the diagnosis of PE. CT pulmonary angiography exhibits at least similar performances, being less invasive and easier to perform. It is nowadays considering as the gold standard.

-Please reduce self-citations, if possible, unless it is absolutely necessary to use previous publications

We have removed the reference on the SPEED application (i.e., Roy PM, Durieux P, Gillaizeau F, et al. A computerized handheld decision-support system to improve pulmonary embolism diagnosis: a randomized trial. Ann Intern Med 2009; 151(10): 677-86.). 

We think that the other references are justified.  

Best regards, 

Prof Pierre-M Roy and Dr Delphine Douillet

Reviewer 3 Report

this narrative review is very interesting .

yet, limitaions to approach PE in daily clinical pratice are major for recurrent PE and this topic is not fully explained, so my suggestion is to add this topic to the review as a separate paragraph. In particular doubtfoul imaging of recurrent PE in the same pulmonary segment and d-dimer values are not useful. As consequence of this the prolonged use of anticoagulation is adopted in these cases with related risks to have side effects.

this the only suggestion that i can do to authors because the high quality of article.

Author Response

We thank the reviewer for this comment. Indeed, there is a risk of over-testing for these patients. However, this is a delicate situation where there is no clear evidence in the literature. We have added this point in the article as below.

“Limited evidence exists for the diagnosis of recurrent venous thromboembolic events. These patients are at risk of over testing: clinical decision rules are less useful and D-dimer performance is reduced. The efficiency of the algorithm is relatively low compared to patients without a history of VTE (Fabiá Valls MJ, van der Hulle T, den Exter PL, Mos ICM, Huisman MV, Klok FA. Performance of a diagnostic algorithm based on a prediction rule, D-dimer and CT-scan for pulmonary embolism in patients with previous venous thromboembolism. A systematic review and meta-analysis. Thromb Haemost. 2015 Feb;113(2):406–13.).”

Best regards, 

Prof Pierre-M Roy and Dr Delphine Douillet

Round 2

Reviewer 1 Report

I thank the authors for the resubmission of the manuscript. However, I still have the same objections, as the authors have not improved the manuscript according to the previous suggestions. 

Author Response

Dear Reviewer, 

We understand your comment. However, we think you are expressing a personal opinion when you state that it is better to do overtesting whatever the cost in order not to miss a diagnosis of pulmonary embolism. Indeed, all authors agree that it is necessary to fight against this overtesting which has many deleterious effects on the patient, the organization of emergency care, and for health insurance (Kahn et al New Engl J Med 2022, Freund et al JAMA 2022, Dobler et al Breathe 2019, Wiener et al Arch Intern Med 2011, Hoda et al BMJ, 2014...).

We recall that the fight against overtesting is done safely as demonstrated by the studies evaluating the new strategies. 

Best regards, 

Prof Pierre-Marie Roy and Dr Delphine Douillet